# In-Orbit Performance of the GRACE Accelerometers and Microwave Ranging Instrument

Michael Murböck [1], Petro Abrykosov [2], Christoph Dahle [3], Markus Hauk [3,4,5], Roland Pail [2] and Frank Flechtner [1,3,*]

1 Institute of Geodesy and Geoinformation Science, Technische Universität Berlin, Straße des 17. Juni 135, 10623 Berlin, Germany

2 Chair of Astronomical and Physical Geodesy, Technical University of Munich (TUM), Arcisstraße 21, 80333 München, Germany

3 Department 1: Geodesy, GFZ German Research Centre for Geosciences, Telegrafenberg, 14473 Potsdam, Germany

4 Max-Planck-Institute for Gravitational Physics (Albert-Einstein-Institute), Leibniz University Hannover, Callinstraße 38, 30167 Hannover, Germany

5 German Aerospace Center (DLR), Institute for Satellite Geodesy and Inertial Sensing, Callinstraße 30b, 30167 Hannover, Germany

* Correspondence: frank.flechtner@gfz-potsdam.de

**Abstract:** The Gravity Recovery and Climate Experiment (GRACE) satellite mission has provided global long-term observations of mass transport in the Earth system with applications in numerous geophysical fields. In this paper, we targeted the in-orbit performance of the GRACE key instruments, the ACCelerometers (ACC) and the MicroWave ranging Instrument (MWI). For the ACC data, we followed a transplant approach analyzing the residual accelerations from transplanted accelerations of one of the two satellites to the other. For the MWI data, we analyzed the post-fit residuals of the monthly GFZ GRACE RL06 solutions with a focus on stationarity. Based on the analyses for the two test years 2007 and 2014, we derived stochastic models for the two instruments and a combined ACC+MWI stochastic model. While all three ACC axes showed worse performance than their preflight specifications, in 2007, a better ACC performance than in 2014 was observed by a factor of 3.6 due to switched-off satellite thermal control. The GRACE MWI noise showed white noise behavior for frequencies above 10 mHz around the level of $1.5 \times 10^{-6} \, \text{m}/\sqrt{\text{Hz}}$. In the combined ACC+MWI noise model, the ACC part dominated the frequencies below 10 mHz, while the MWI part dominated above 10 mHz. We applied the combined ACC+MWI stochastic models for 2007 and 2014 to the monthly GFZ GRACE RL06 processing. This improved the formal errors and resulted in a comparable noise level of the estimated gravity field parameters. Furthermore, the need for co-estimating empirical parameters was reduced.

**Keywords:** GRACE; accelerometer transplant; microwave ranging instrument post-fit residuals; stochastic modeling; monthly gravity field determination; empirical parameters

## 1. Introduction

The Gravity Recovery and Climate Experiment (GRACE, 2002–2017) [1] satellite mission has, for the first time, provided global observations of seasonal, inter-annual and long-term variations in Terrestrial Water Storage (TWS) for large- and medium-size hydrological catchments, down to an area of about 200,000 km² [2]. This innovative source of information on water fluxes has allowed for improving, e.g., hydrological modeling (e.g., [3–5]) or determining changes in the mass of ice sheets (e.g., [6,7]). TWS, as observed directly by satellite gravity missions, is also of unprecedented value for the assessment of climate variability and as a constraint for state-of-the-art climate models (e.g., [8]). Therefore, the recent Global Climate Observing System (GCOS) implementation plan [9] has defined TWS as an official GCOS Essential Climate Variable (ECV).

The GRACE and GRACE-FO (GRACE Follow-on, since 2018) [10] data, with their key observations of Satellite-to-Satellite Tracking (SST) in low–low and high–low mode, contain indirect information about the Earth's gravitational field. From the different approaches to inferring global gravity field models from these observations, the so-called "classical" or "dynamic" approach [11] is used by the German Research Center for Geosciences (GFZ) [12], the Center for Space Research (CSR) [13] at the University of Texas at Austin, NASA's Jet Propulsion Laboratory [14], as well as by the Technical University of Graz [15]. Most of these gravity field solutions use global Spherical Harmonic (SH) base functions and solve for unknown SH coefficients at monthly time intervals. In order to assess the effects of different error sources and the benefit of specific observation techniques, numerical simulations are widely used. These simulations apply similar processing schemes and theoretical approaches, as described above, for various gravity field mission concepts (e.g., [16–19]). For such simulations, it is essential to apply noise information that is as realistic as possible, as well as corresponding stochastic models for the key observations, and an adequate parameterization scheme.

After analyzing 15 years of GRACE and more than 4 years of GRACE-FO data, there is still high potential to improve processing methods further, in order to extract the signal content of the satellite instrument data to the best possible extent. One potential field of research during gravity field estimation is the stochastic modeling of observation uncertainties. Ref [20] discusses the procedure for fitting a stochastic model for the different observation groups used for [15,21]. These play a crucial role in parameter adjustment because the involved instruments typically have a correlated error behavior to be introduced as a metric into the parameter estimation process. Additionally, since different types of observations are combined, the scaling of the stochastic model provides a relative weighting among observation groups. Even though pre-launch estimates for the critical payload exist, the stochastic in-orbit behavior of the observation noise is unknown. This study provides an innovative analysis of the GRACE SST and accelerometry observations and their systematic error behavior. This analysis is used to derive an approximation of corresponding stochastic models, with a focus on potential changes of the instrument characteristics over time. Without adequate stochastic modeling, the co-estimation of dedicated empirical parameters is required to reduce the impact of incorrectly modeled forces [12]. In [22], it was shown in a closed-loop simulation environment that, by introducing a realistic stochastic model for the accelerometers, the need for the co-estimation of empirical parameters can be avoided. Hence, additional correlations between parameter groups can also be reduced.

To improve the separation of signal and noise in the estimated gravity field parameters, we targeted the stochastic of the GRACE key instruments, i.e., the ACCelerometers (ACC) and the inter-satellite MicroWave ranging Instrument (MWI). The GRACE ACC data feature highly correlated noise and different types of systematic effects (e.g., thruster and heater activations), which have to be considered [23–26]. For the ACC data, we applied a "transplant approach", where the residuals of transplanted accelerations from one GRACE satellite to the other were analyzed. Similar approaches were applied to GRACE and GRACE-FO ACC data to overcome the deficiencies of single-degraded ACC observations at the end of the GRACE mission lifetime [27] or throughout the GRACE-FO mission [28].

To retrieve the noise in the MWI data, we analyzed different types of residuals resulting from the monthly GFZ GRACE RL06 Level-2 processing [12]. Besides gravitational and non-gravitational signals, the MWI data consist of sensor noise and other error effects, which were analyzed in [29] based on inter-satellite accelerations. Other studies focused on the MWI system noise and revealed error effects due to temperature variations and satellite eclipse crossings [30,31]. For the GRACE-FO data [32], the observations from the two ranging instruments, the MWI and the Laser Ranging Instrument, should be compared, and dependencies found on the carrier frequency.

In contrast to our approach for MWI data, the method for retrieving the ACC noise is not suitable for GRACE-FO data, as the ACC performance on one of the two GRACE-FO satellites is heavily degraded. Therefore, specific ACC transplant approaches are proposed

for deriving ACC data of comparable quality for the degraded instrument [27,28]. Hence, our ACC approach did not result in reliable estimates for the ACC noise, as it assumed independent but comparable noise characteristics on the two satellites. Therefore, as the title states, this study focused on GRACE data, and the application to GRACE-FO is the subject of future work.

In this study, we present instrument performance analyses for the GRACE MWI and ACCs, from which stochastic models for the estimation process can be derived. The corresponding research questions are:

- How can the errors of the GRACE key instruments (ACC and MWI) be deduced?
- How do those instruments behave systematically and stochastically?
- What is the impact of using a combined ACC+MWI stochastic model on GRACE time variable gravity field determination?

In Sections 2 and 3, we describe the methods and results of the performance analyses of the GRACE key instruments (ACCs and MWI). Section 4 contains an analysis of the impact that the application of the stochastic models for the combined effects of ACC and MWI errors have on gravity field determination.

## 2. Performance Analysis Methods

In this section, we describe the methods for retrieving the stochastic properties of the GRACE key instruments, i.e., the ACCs and the MWI, based on the corresponding Level-1B (L1B) data.

### 2.1. Accelerometers

The twin GRACE satellites fly on near-identical orbits at an initial altitude of 500 km with an average inter-satellite distance of 220 km. It can, therefore, be assumed that the trailing satellite's (A) ACC observes nearly the same non-gravitational forces (caused by air drag, solar radiation pressure and Earth albedo) as the leading satellite (B) but with a specific temporal delay. This temporal delay depends on the actual inter-satellite distance and constitutes approximately 25 s. Hence, we assumed that the ACC noise dominates the remaining differences of the non-gravitational forces in the relevant frequencies. Since both GRACE satellites carry the same SuperSTAR-type ACC [33], their performance respective noise properties can be considered identical. Subtracting the time-shifted ACC time series of satellite A from that of satellite B in each spatial direction would then cancel out the signal, leaving a time series that consists of the noise of both satellites. Both ACCs are subject to disturbing effects, which can occur either in a regular pattern (e.g., temperature fluctuations due to switching of on-board electronics) or randomly (e.g., thruster firings for orbital control). Since these disturbing effects are unique for each satellite, they have to be considered accordingly before subtraction.

Required L1B (release 2.0) data products:

- ACC1B: 1 s sampling; ACC measurements in X (along-track), Z (radial) and Y (completes a right-handed system with X and Z, cross-track).
- GPS Navigation L1B (GNV1B): 5 s sampling; orbital positions of each satellite in an earth-fixed frame.
- Thruster L1B (THR1B): irregular sampling; contains time stamps and duration of thruster firings.
- Star Camera Assembly L1B (SCA1B): 5 s sampling; orientation of the satellites in terms of quaternions in an inertial frame.

Individual steps of the ACC transplant processing strategy:

1. GNV1B and SCA1B sampling is increased using linear interpolation to match that of ACC1B. Then, the time shift that separates both satellites is determined based on the minimal distance of orbital positions. This time shift can then be applied to either satellite's data products (with the respective sign). We applied it to those of satellite B.

2.  Data points contaminated by non-stochastic effects, i.e., thruster firings—the only deterministic error component provided in the L1B data—are removed. Although thruster firings may occur in a rapid sequence, each firing is generally short-periodic (well below one second). In the ACC Level-1A (L1A) data (raw 10.034 Hz ACC readings), they also appear as such. The processed ACC1B results from, amongst others, applying a 35 mHz filter to the L1A data. Therefore, these short pulses are smeared over periods of over 70 s (e.g., [27]). This error source is removed from the ACC1B data by cutting all data points in the range of 40 s to both sides of every thruster firing's time stamp.
3.  ACC1B data, which are flagged with "proof mass voltage out of nominal range", can be considered as irreversibly compromised and are, therefore, cut from the time series of both satellites. It is noted that within the frame of the analyses performed for this study, cutting data with other types of flags did not yield a better overall result.
4.  Remaining outliers in the ACC1B time series exceeding a 3-sigma criterion are removed.
5.  To achieve an optimal fit between the homogenized ACC1B-A/B time series, a least-squares adjustment was carried out where a scale factor and a six-parameter polynomial (bias, linear drift and periodic components of 1 and 2 Cycles Per Revolution (CPR)) was estimated daily for every ACC axis (cf. Equation (1)). Additionally, we considered the satellites' relative orientations at their evaluated orbital positions (derived from SCA1B-A/B). It should be emphasized that we did not estimate the absolute bias, drift and scale but rather the relative values (i.e., one of the scale factors is fixed to 1). The observation equation for satellites *A* and *B* can be written as follows:

$$
\begin{aligned}
&\underline{\Delta \hat{S}}_A \underline{a}_A + \underline{\Delta \hat{Q}}_A + \underline{\Delta \hat{D}}_A t + \underline{\hat{A}}_{1,A} \cos \omega_1 t + \underline{\hat{B}}_{1,A} \sin \omega_1 t + \underline{\hat{A}}_{2,A} \cos 2\omega_1 t + \underline{\hat{B}}_{2,A} \sin 2\omega_1 t \\
&= R_A^B \left( \underline{\Delta \hat{S}}_B \underline{a}_B + \underline{\Delta \hat{Q}}_B + \underline{\Delta \hat{D}}_B t + \underline{\hat{A}}_{1,B} \cos \omega_1 t + \underline{\hat{B}}_{1,B} \sin \omega_1 t + \underline{\hat{A}}_{2,B} \cos 2\omega_1 t + \underline{\hat{B}}_{2,B} \sin 2\omega_1 t \right)
\end{aligned}
\tag{1}
$$

where

| | |
|---|---|
| $\underline{\Delta \hat{S}}_{A/B}$ | relative scale factors (estimated daily) |
| $\underline{\Delta \hat{Q}}_{A/B}$ | bias (estimated daily) |
| $\underline{\Delta \hat{D}}_{A/B}$ | linear drift (estimated daily) |
| $\underline{\hat{A}}_{1/2,\,A/B}, \underline{\hat{B}}_{1/2,A/B}$ | 1/2 CPR signal component scale factors (estimated daily) |
| $R_A^B$ | Rotation matrix (derived from SCA1B) |
| $\underline{a}_{A/B}$ | ACC1B observations |
| $t$ | time |
| $\omega_1$ | angular velocity corresponding to 1 CPR |

The rotation matrix $R_A^B$ is derived from two rotations computed from the SCA1B data for the two satellites, i.e., from the satellite reference frame (SRF) of satellite B to the inertial frame, and to the SRF of satellite A. Solving Equation (1) for $\underline{a}_A$ and $\underline{a}_B$ then yields the functional model for the adjustment (both ACC1B-A and -B are regarded as observations). Finally, the adjusted ACC1B time series for X, Y and Z (left and right side of Equation (1)) are subtracted from each other. Then, the square-root of the Power Spectral Density (PSD), i.e., the Amplitude Spectral Density (ASD), of these ACC residuals, which is now assumed to be comprised exclusively of instrument noise, is computed.

### 2.2. Microwave Instrument

The GRACE MWI range rate noise is dominated by two components, the system and the oscillator noise. The former consists of thermal noise caused by the K-Band system itself, and in terms of ranges, it dominates the high frequencies above approx. 1 mHz with a white noise level of $10^{-6}$ m/$\sqrt{\text{Hz}}$. The latter dominates the low frequencies with an $f^{-1}$ behavior [16]. We focused on range rate K-Band Ranging (KBR) L1B data, which are also used for the current GFZ RL06 Level-2 (L2) processing [12]. As it is not necessary to estimate K-Band instrument

parameters within L2 processing [12], it can be assumed that for this observation type, no bias, jumps and trends occur for large parts of the GRACE mission duration.

Nevertheless, when using these data for gravity field determination, it is recommended to look for outliers or other special events such as maneuvers or reboots of the instrument processing unit, as performed, e.g., for GFZ GRACE RL06 processing. Furthermore, as for GFZ GRACE RL06 L2 processing, we processed single arcs of a maximum one-day length, while all effects featuring longer periods were neglected. In addition to gravity parameters in terms of SH coefficients up to maximum degrees of 60 or 96, we estimated different empirical and instrumental parameters such as periodic once-per-revolution accelerations, and ACC biases and scales. Therefore, these parameter groups reduced some error effects from different sources. We analyzed the KBR1B observations and other types of residuals within the GFZ RL06 L2 processing, in order to investigate different signal and noise content in the data. Three different types of residuals were considered (more details in [12]):

- "Editing residuals": calculated before gravity field modeling and used for MWI data editing; L1B data reduced by an a priori static and time-variable gravity field signal and by other a priori signals, e.g., tidal signals.
- "Pre-fit residuals": calculated after data editing and used for gravity field modeling; L1B data reduced by an a priori static gravity field signal and by other a priori signals, e.g., tidal signals (resulting in usually larger amplitudes than the editing residuals in the frequency band between 1 and 10 mHz).
- "Post-fit residuals": calculated after gravity field modeling; L1B data reduced by all best-known estimated parameters.

As an example, Figure 1 shows the ASDs of the three residual types for the MWI range rate data of 1 January 2007. Frequencies above 10 mHz are dominated by the MWI instrument noise. Therefore, all three types show very similar behavior in this spectral band. The frequencies below 10 mHz are dominated by other noise sources (e.g., ACCs) and residual gravitational signals (e.g., from high-frequency atmospheric and oceanic mass variations). Here, the three residual types show different spectra. For the case that no a priori time-variable gravity field signal is subtracted, the editing residuals (blue) show smaller amplitudes than the pre-fit residuals (red, frequency range between 0.6 and 10 mHz), and the post-fit residuals (yellow) show the smallest amplitude spectrum. The co-estimation of empirical parameters such as once per rev. amplitudes reduces the ASDs for frequencies below 0.5 mHz, which becomes apparent when compared to the residual ASDs of the solutions where no empirical parameters are estimated.

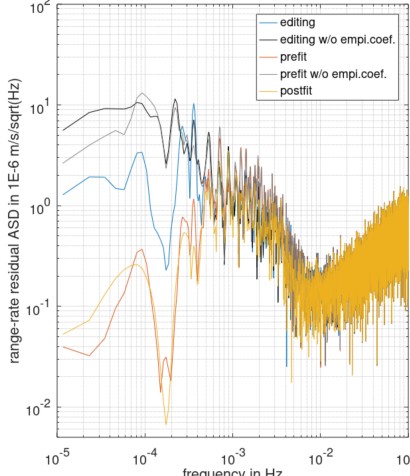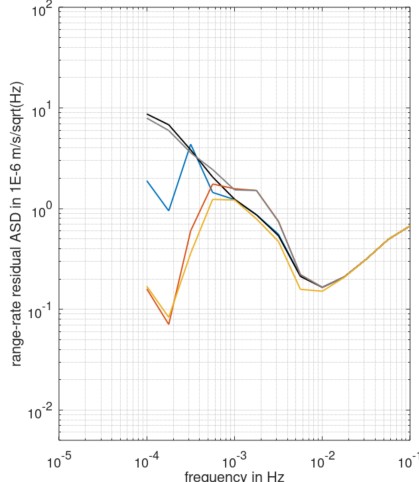

**Figure 1.** GRACE MWI range rate ASDs for 1 January 2007 for editing (blue), pre-fit (red) and post-fit (yellow) residuals with setting up empirical parameters, as well as editing and pre-fit residuals without setting up empirical parameters (black and gray). (**Left**) ASD using a 1-day Hann window [34]. (**Right**) smoothed mean ASD for $10^{\{-4.00, -3.75, -3.50, \ldots -1.00\}}$ Hz.

## 3. Results

In this section, the results of the performance analyses of the GRACE key instruments (ACCs and MWI) are discussed. The primary input data are the GRACE L1B products for the test years 2007 and 2014. Hence, the data were analyzed in different phases of the GRACE mission at different altitudes (nearly 470 km in 2007 and around 420 km in 2014), with different data quality (e.g., activated/deactivated thermal control in 2007/2014) and different environmental conditions (low solar activity in 2007 and high solar activity in 2014).

### 3.1. Accelerometers

The procedure described in Section 2.1 was carried out for every day of the years 2007 and 2014 (as far as ACC1B data were available). The results are shown in Figure 2 in terms of monthly and annual mean ASDs. It can be seen that the noise level of each measurement axis (especially for the sensitive axes X and Z) remains stable over one year. For 2007, for the sensitive axes X and Z, the minimum and maximum monthly ASD values between 0.1 and 10 mHz range from 0.5 to 1.5 of the annual mean ASD. For 2014, these values range from 0.7 to 1.5 of the annual mean. The results also imply that the estimation of daily bias, scale and linear drift is sufficient. The ACC measurements in the less-sensitive Y-axis are around one order of magnitude above the pre-launch noise specifications (e.g., [23]). The X and Z measurements, however, notably deviate from their specifications in the spectral range relevant for gravity field retrieval (between 0.1 and 10 mHz) by two orders of magnitude. These deviations are more substantial in 2014 than in 2007. No definitive statement towards the alignment of the noise model and the instruments' true performance can be drawn for the high-frequency range, since the impact of the low-pass filtering (cut-off at 35 mHz), carried out in the L1B processing, is predominant.

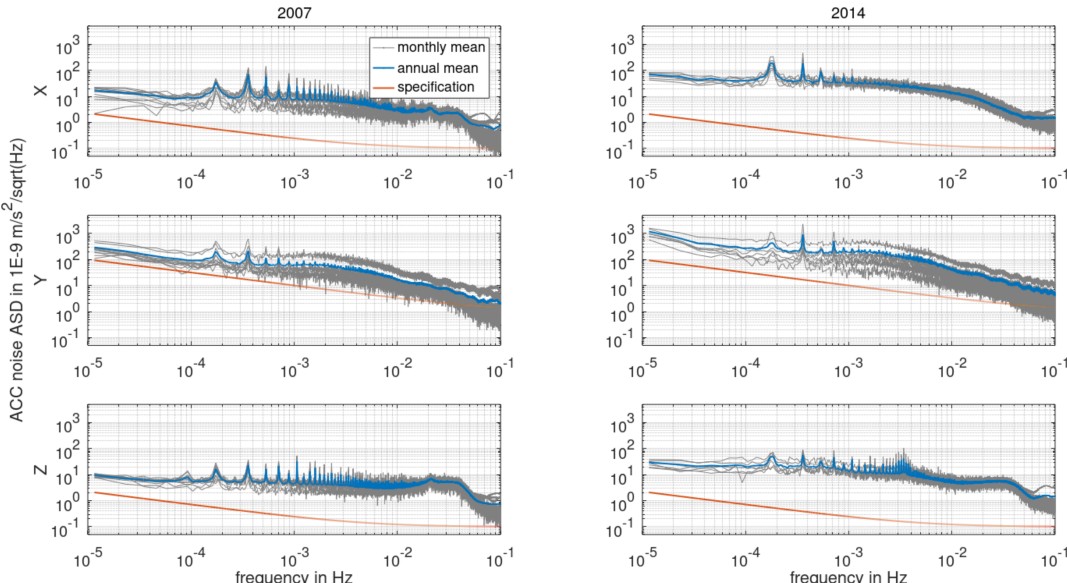

**Figure 2.** Monthly (gray) and annual (blue) mean noise ASDs of the GRACE ACCs for 2007 (**left**) and 2014 (**right**) for all three axes in the satellite reference frame in comparison with the ACC noise specifications (red).

Instead, the most likely underlying cause of the ACC measurements not meeting their specifications was found to be the ACC1A to ACC1B data processing. To better understand the cause of the deviations, additional studies of L1A data were carried out. L1A features raw 10.034 Hz ACC measurements, which encompass a multitude of undamped high-frequency disturbing effects. These effects include the formerly mentioned thruster firings, twangs (high amplitude and -frequency oscillations of speculative origin, e.g., [24]),

and fluctuations due to the switching of heaters, magnetotorquers and other on-board instruments. It is noted that the thruster firings only constitute a small part of all disturbing effects.

However, the ACC1A data features some short time spans of up to a few minutes duration, in which the measurements are not affected by any disturbances (usually not simultaneously for both satellites). For such time spans, the instrument noise can be approximated by reducing a linear and, if necessary, a quadratic trend from the time series. This approach was used in [23], where it was found that in these specific time spans the ACC1A noise is indeed widely in accordance with its pre-flight specification. The ACC1B data, on the other hand, do not feature such error-free intervals, or they are far too short to estimate a meaningful PSD, because the low-pass filtering smears any disturbing signals over time intervals of ca. 70 s.

Due to the irreversible nature of this smearing, these components cannot be separated on L1B level. This would only be feasible with a more sophisticated ACC1A to ACC1B processing which specifically targets and removes the ACC disturbances before the low-pass filter is applied. While such a strategy is possible in principle (e.g., [23], the modeling of heater spikes, and [24], twang modeling), it exceeds the scope of this study and is, therefore, left as a recommendation. It is shown that the retrievability of the actual ACC noise on L1B level is limited, and a dedicated pre-processing of ACC1A data is suggested. Nevertheless, because the ACC1B data are used in gravity field processing as is, it is still reasonable to derive a stochastic model based on the results shown in Figure 2.

Further on, as we assume stationary ACC noise over the course of one year, a single variance covariance matrix (VCM) for each test year is sufficient. The systematic effects to be accounted for are found to be the ACC bias, scale and linear drift for each axis. The fact that daily solutions yield noise ASDs of slightly lower levels than the monthly solutions indicates that these three factors show variations over the period of one month, and a daily estimation is, therefore, recommended.

### 3.2. Microwave Instrument

The objective of this section is the characterization and deduction of stochastic models for the GRACE MWI range rate noise, applying the methods described in Section 2.2. The main outcomes are based on analyses of L1B data and residuals within the GFZ GRACE RL06 gravity field L2 processing [12]. The analyses were performed in the frequency domain and in comparisons with pre-launch instrument specification models. The resulting stochastic models in terms of ASDs are discussed.

To derive a stochastic model for the GRACE MWI observations, long period analyses were performed for range rate post-fit residuals. From the estimated ASDs, we also computed smoothed ASD values for the ACC data from the previous section and for the MWI data for the frequencies $10^{\{-4, -3.75, -3.5, \cdots\}}$ Hz, in order to quantify the different noise levels. Figure 3 shows these smoothed mean ASDs for 2007 and 2014 in comparison to the MWI requirements ([35]; the specifications for the GRACE-FO instruments are assumed to be the same as for GRACE) and the mean annual ACC noise ASDs. The high-frequency spectrum (above 10 mHz) is very similar for both years and shows the same increase with increasing frequency. Mainly due to some extreme values in December 2007, the mean smoothed ASD values above 10 mHz are approx. 20% larger than in 2014.

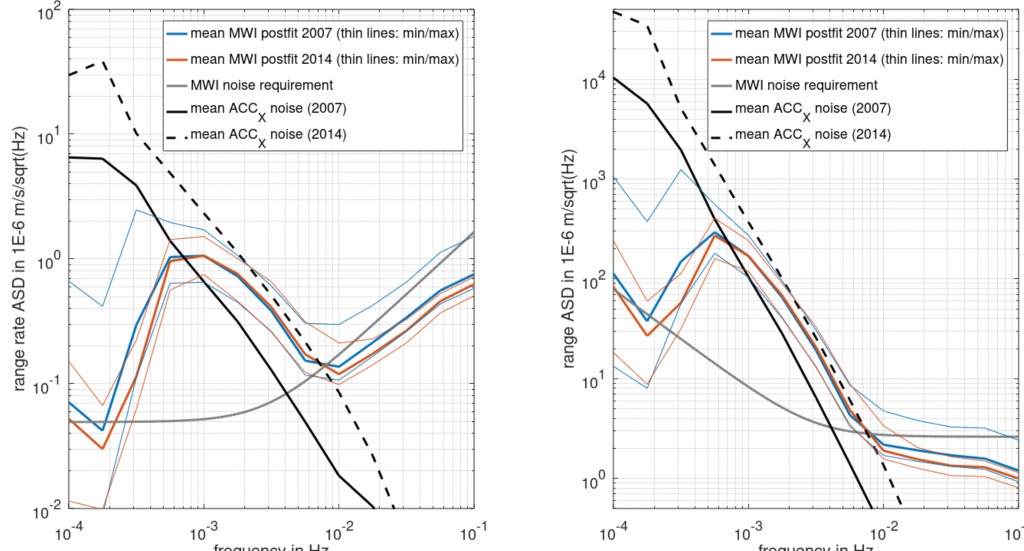

**Figure 3.** Mean annual smoothed MWI post-fit residual ASDs for 2007 (blue, thick lines are the mean; thin lines the min/max) and 2014 (orange) in terms of range rates (**left**) and ranges (**right**), in comparison with the MWI noise requirements and the mean along-track ACCX noise for 2007 (solid black, cf. Figure 4, left) and 2014 (dashed black).

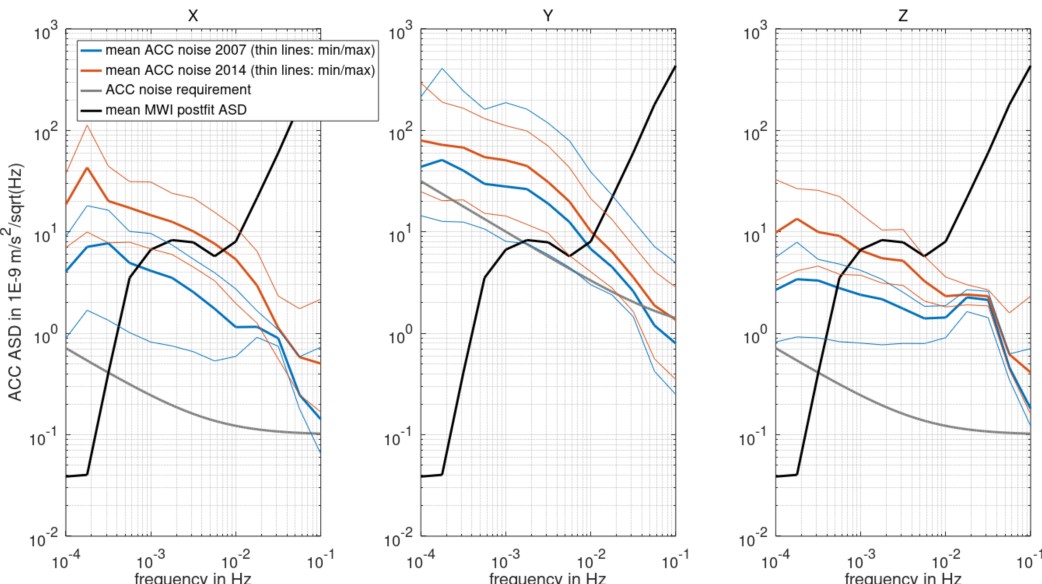

**Figure 4.** Mean annual smoothed noise ASDs of the GRACE ACCs for 2007 (blue, thick lines are the mean; thin lines the min/max) and 2014 (red) for X (**left**), Y (**middle**), Z (**right**), in comparison with the ACC noise specifications (gray) and the mean MWI post-fit ASDs (black, cf. Figure 3).

To assess a stochastic model for the GRACE MWI ranging observations, we focused on the high frequencies because the low frequencies include a number of other effects (e.g., ACC noise) which do not feature a stochastic behavior. When transforming the range rate residual ASDs to ranges (down-scaling by the factor $2\pi f$, with f being the frequency range), a near-white noise behavior can be assessed in the high frequencies. The smoothed MWI post-fit ASD values for 2007 are shown in Figure 5 (top) for frequencies above 10 mHz in terms of time series. They vary from 2 to more than $4 \times 10^{-6}$ m/$\sqrt{\text{Hz}}$ for the 10 mHz values (light gray) and from 1 to $2 \times 10^{-6}$ m/$\sqrt{\text{Hz}}$ for the 100 mHz values (black). The variations for this spectral range are partially related to the inter-satellite distance but not to the ACC noise. Other reasons for the different variations (e.g., the sharp MWI post-fit ASD

maximum in December) are discussed in [30,31]. For comparison, this plot also contains the ACC noise time series for the three axes for 10 mHz. With the exception of the less sensitive Y-axis, there are hardly any variations, and the absolute values are around one order of magnitude smaller than the MWI noise. Hence, the range rate noise above 10 mHz is not dominated by the ACC but mainly by the MWI.

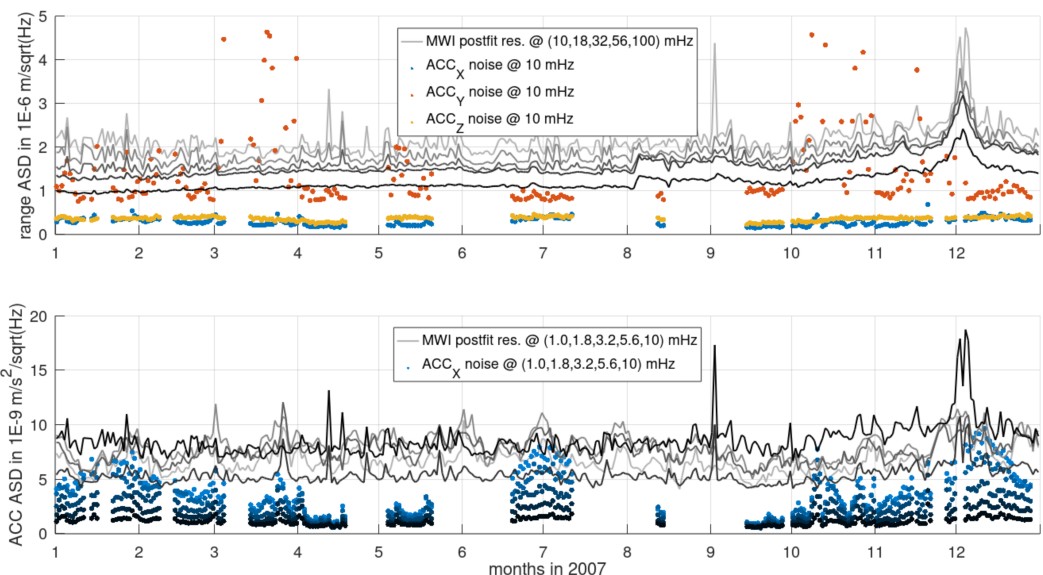

**Figure 5.** Smoothed ASD values for range rate post-fit residuals (gray) and ACC noise (colored) for 2007 for different frequency bands. (**Top**) High frequencies (between 10 and 100 mHz) in terms of ranges comparing the range rate post-fit residuals with the three ACC axes for 10 mHz. (**Bottom**) medium frequencies (between 1 and 10 mHz) in terms of accelerations comparing the range rate post-fit residuals with the X-axis ACC noise.

For the frequency part below 10 mHz the situation is different. In Figure 4, the mean smoothed ACC ASDs for 2007 (blue) and 2014 (red) are shown in comparison to their specifications (gray) and the mean smoothed range rate post-fit ASD (black). In general, the ACC noise is much larger than the range rate post-fit residuals for frequencies below 1 mHz, mainly due to the co-estimation of empirical parameters within GFZ L2 processing. Between 1 and 10 mHz, the different ACC noise ASDs are at the same level as the range rate residuals. We assumed that the high noise from the less sensitive cross-track ACC Y-axis would not affect the range rate post-fit residuals. The noise from the radial Z-axis seems to be significantly smaller than for the along-track X-axis and the range rate post-fit residuals in this frequency band. Taking a closer look at the along-track X-axis in this frequency band, Figure 5 (bottom) shows the smoothed range rate post-fit and the smoothed ACC noise ASD values for this frequency band in terms of a time series for 2007. In particular, the data for 1 mHz (light gray vs. light blue) indicate that the range rate post-fit residuals are partly dominated by the X-axis ACC noise (increase in January, July and December).

Summarizing the results for the MWI noise, we observed a constant behavior for high frequencies (above 10 mHz) when neglecting smaller variations between 1 and $3 \times 10^{-6}$ m/$\sqrt{\text{Hz}}$. This resulted in a stationary noise ASD model for MWI observations, which can be approximated by the analytical models (Table 1 and Figure 6, left).

$$\delta_{\text{GRACE–MWI}} = 1.5 \times 10^{-6} \times \sqrt{1 + (10 \text{ mHz}/f)^2} \text{ m}/\sqrt{\text{Hz}} \tag{2}$$

Analyzing the GRACE-FO range rate post-fit residuals revealed that the GRACE-FO MWI performs better than the GRACE MWI by more than a factor of 2. For the high

frequencies, the range noise ASDs are approximated by $1.5 \times 10^{-6}$ m/$\sqrt{\text{Hz}}$ (Equation (2)) for the GRACE MWI and by $6.3 \times 10^{-7}$ m/$\sqrt{\text{Hz}}$ for the GRACE-FO MWI.

For both test years and all three ACC axes, we see a significantly higher noise than their specifications. However, for X-axis frequencies below 10 mHz, the 2007 performance is approximately three times better than for 2014. In these frequencies, the ACC noise on GRACE dominates that of the MWI, and the corresponding ASD can be approximated by the analytical models $1.3 \times 10^{-10}/\sqrt{f}$ m/s$^2$/$\sqrt{\text{Hz}}$ for 2007 and $4.7 \times 10^{-10}/\sqrt{f}$ m/s$^2$/$\sqrt{\text{Hz}}$ for 2014 (Table 1 and Figure 6, right).

**Table 1.** Analytical ACC noise ASD parameters a for a model $a/\sqrt{f}$ (frequencies below 10 mHz) in $10^{-10}$m/s$^2$, and the MWI noise parameters b for a constant model (frequencies above 10 mHz) in $10^{-6}$ m.

|  | ACC$_X$ | ACC$_Y$ | ACC$_Z$ | MWI |
|---|---|---|---|---|
| 2007 | 1.3 | 8.2 | 1.0 | 1.7 |
| 2014 | 4.7 | 13 | 2.4 | 1.4 |

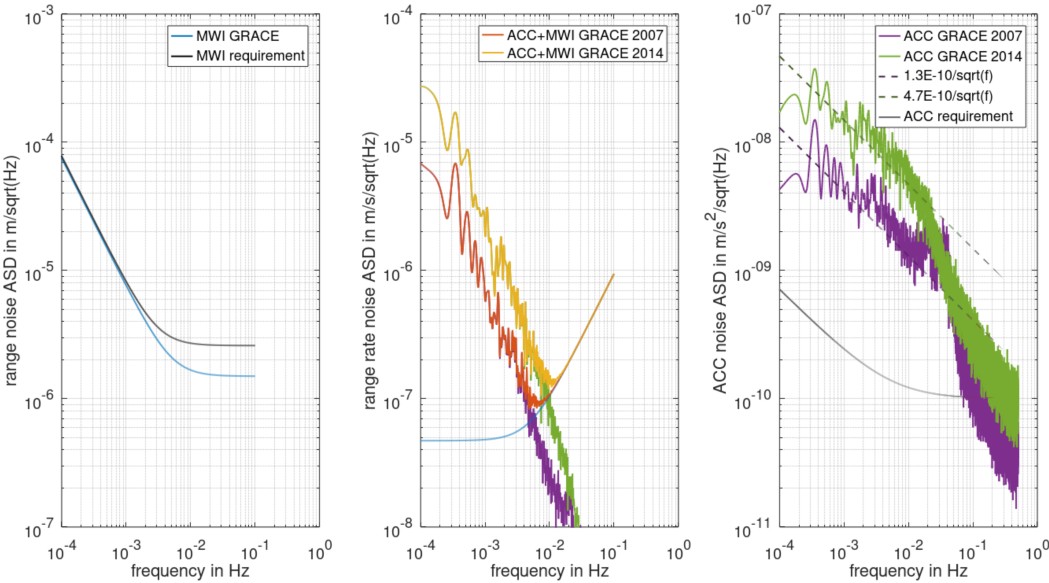

**Figure 6.** Noise models in terms of ASDs of the key GRACE instruments. (**Left**) GRACE MWI (blue) noise model and requirements in terms of range ASDs in m/$\sqrt{\text{Hz}}$ (cf. Equation (2)). (**Right**) GRACE ACC noise models for 2007 (purple) and 2014 (green) with requirements in terms of acceleration ASDs in m/s$^2$/$\sqrt{\text{Hz}}$. (**Center**) combined GRACE ACC+MWI noise models in terms of range rate ASDs in m/s/$\sqrt{\text{Hz}}$ (cf. Equation (3)).

## 4. Impact on Gravity Field Retrieval Performance

Combining the results from Sections 3.1 and 3.2, representations for combined stochastic models representing the ACC and MWI noise are shown in Figure 6. The MWI noise (left figure in terms of ranges) dominates the high frequencies, while the ACC noise (right figure in terms of accelerations) dominates the low frequencies of the range rate observations (middle figure in terms of range rates). For the MWI noise, we used the stationary analytical model and for the ACC noise, we used the mean ASDs for the along-track component for the two years, resulting in two combined ACC+MWI noise models for 2007 and 2014. Let $\delta_{\text{GRACE}-\text{KRA}}$ (2) be the MWI noise model in terms of range ASDs and $\delta_{\text{GRACE}-\text{ACC}-2007/2014}$ the ACC noise model, as shown in Figure 6 (right), in terms of acceleration ASDs; then, the combined noise model in terms of range rates (Figure 6, center) is

$$\delta_{\text{GRACE}-\text{ACC}+\text{MWI}-2007/2014} = \sqrt{(\delta_{\text{GRACE}-\text{KRA}} \times 2\pi f)^2 + (\delta_{\text{GRACE}-\text{ACC}-2007/2014}/2\pi f)^2}. \qquad (3)$$

For this spectral representation, the required derivative/integration in the time domain is performed by scaling by $2\pi f^{\pm 1}$, respectively.

Within gravity field adjustment, we estimated the gravity field parameters from the range rate observations, which were reduced by the non-gravitational effects deduced from the ACC observations. Hence, the range rate observations contain the noise from two instrument types, i.e., the MWI and the ACC. Assuming a stationary process, the 24-h auto-covariance function derived from the combined ASD (Figure 6, center) fully describes the stochastic process from these noise contributions. Following [20], the auto-covariance function fills the full 24 h VCM with a Toeplitz structure. To apply this stochastic model in terms of a decorrelation, the Cholesky-decomposed inverse of this VCM (lower triangular) is applied to the observations and the observation equations in the design matrix. We used this more direct way of applying the stochastic model compared to the application of an auto-regressive moving average filter, as used in [36].

The characteristics of the filter matrices, e.g., the effects of the filter length, and of different ACC noise characteristics, were assessed. The filter length was tested for periods between 3 and 24 h by replacing the filter matrix entries, which are more than, e.g., 3 h apart from the main diagonal, with zeros. As a filter length of 24 h for some months resulted in degraded solutions and no difference in performance regarding the formal errors of the estimated gravity field parameters was observed, a filter length of 3 h was chosen for the long period tests (cf. Figure 7, left). For the ACC noise model, we tested filters which neglect (as well as include) the peaks in the ACC noise ASD around multiples of the orbital frequency (approx. 0.18 mHz). While here, the performance of the estimated gravity field parameters did not change, the formal error behavior resulted in much more realistic values when including the peaks in the ACC error model. Therefore, this more realistic error model was chosen for the long period tests.

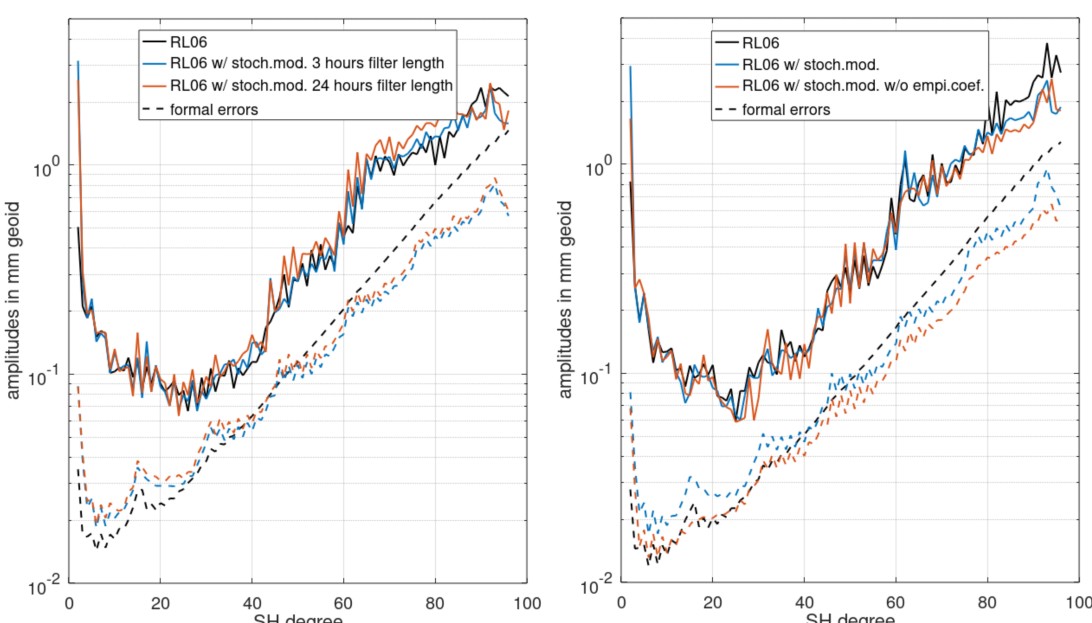

**Figure 7.** Monthly GRACE solutions for November 2007 (**left**) and January 2007 (**right**) in terms of SH degree amplitudes in mm geoid heights of differences with respect to a GFZ GRACE RL06 climatology model (solid lines), and of formal errors (dashed lines), comparing RL06, RL06 applying stochastic modeling with different filter lengths (**left**), and RL06 with stochastic modeling (3 h filter length) with and without estimating empirical parameters (**right**).

In order to assess the impact of the improved stochastic modeling of ACC and MWI instrument data, we implemented, validated and tested these stochastic models for real data applications. The effect on the GFZ monthly GRACE L2 processing is depicted in

Figure 8 in terms of formal errors and residuals with respect to a GRACE climatology model. The GRACE climatology model consists of six parameters estimated for each SH coefficient $K$ (any $C_{nm}$ and $S_{nm}$ of degree n and order m) from the whole GFZ GRACE RL06 time series. It reads

$$K(t) + e(t) = a + bt + c_a \cos(2\pi t) + s_a \sin(2\pi t) + c_s \cos(4\pi t) + s_s \sin(4\pi t) \quad (4)$$

where

| | |
|---|---|
| $t$ | Time in years |
| $K(t) + e(t)$ | SH coefficient time series with residuals |
| $a$ | Constant parameter |
| $b$ | Linear parameter |
| $c_{a/s}$, $s_{a/s}$ | Cosine/Sine amplitudes of the annual/semi-annual signal. |

The results show more realistic formal errors, i.e., a posteriori formal standard deviations of the estimated parameters. These show higher amplitudes around SH 15th-order resonances (15, 30, etc.) and, therefore, better follow the characteristics of the residual RMS SH degree amplitudes. The residual RMS for the two years is increased for medium SH degrees (around 50), especially for near-sectorials, and decreased for low (<30) and high (>70) SH degrees. On average for the two years 2007 and 2014, the latitude-weighted ocean RMS of the 250 km Gaussian-filtered residuals with respect to the GRACE climatology model (Equation (4)) (excluding a 500 km buffer zone around land areas) were reduced by 2% (standard deviation 7%), but also increased by 8% for 400 km Gaussian filtering (standard deviation 9%) (cf. Table 2 for January 2007). In the filtered cases, the amplitudes of the formal errors are significantly larger in 2014 than in 2007, which may be related to the higher ACC noise in 2014. These results already show the need for further optimized stochastic modeling in combination with the relative weighting of the KBR and the Global Positioning System (GPS) part of the least squares system.

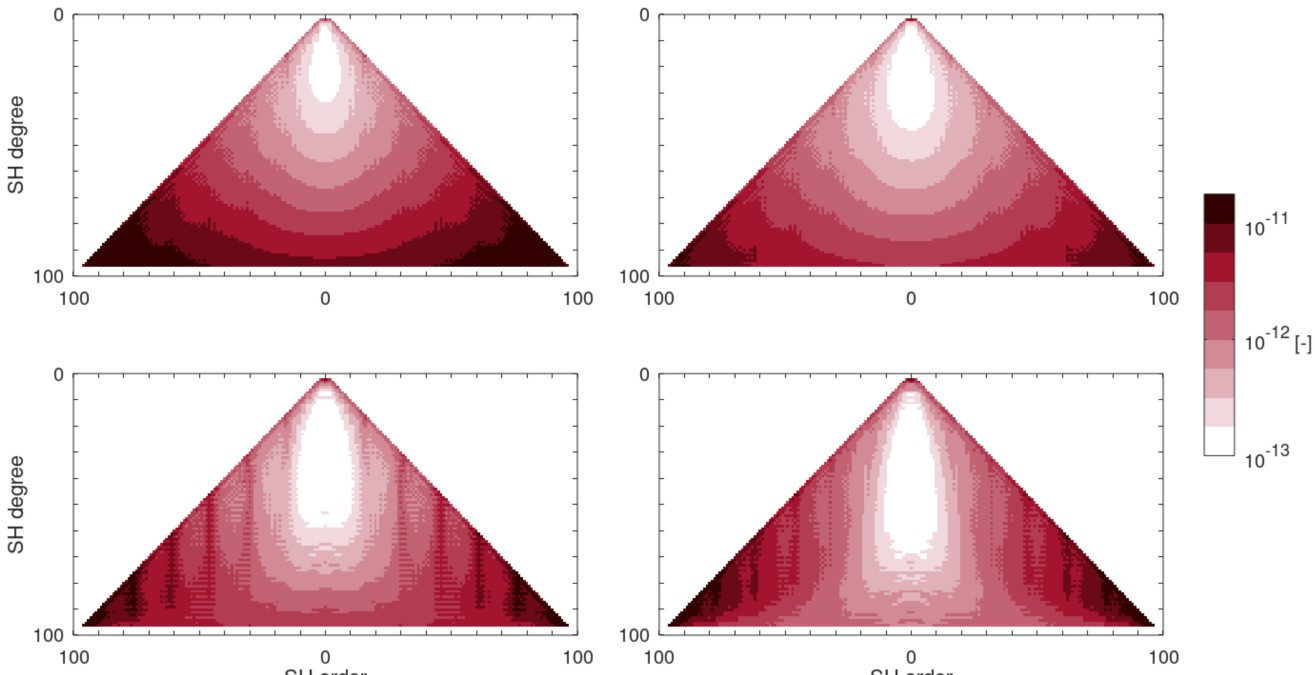

**Figure 8.** *Cont.*

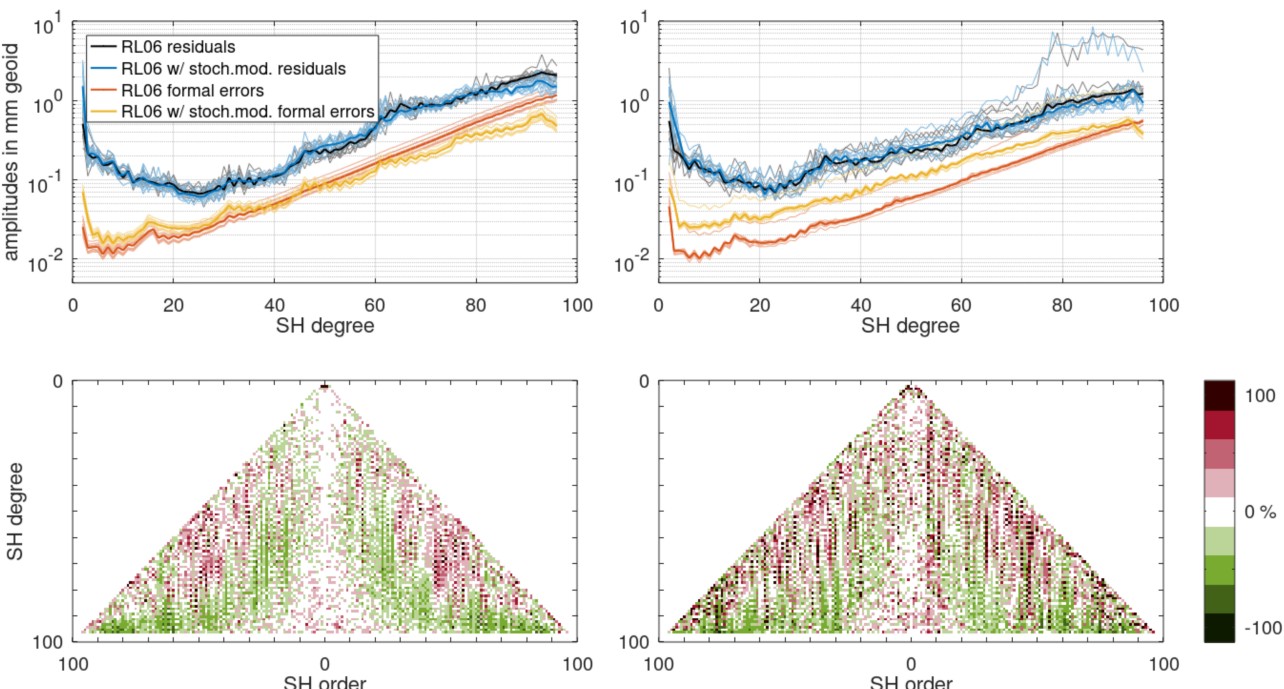

**Figure 8.** Comparison of the standard GRACE GFZ RL06 solution and a solution with the application of the combined ACC and MWI a priori stochastic model for 2007 (**left**) and 2014 (**right**). (**Row 1 and 2**) Logarithmic scaled formal error median of all monthly solutions for the standard RL06 solutions (**Row 1**) and those with the a priori stochastic model applied (**Row 2**). (**Row 3**) Median of SH degree amplitudes (thick lines) of the monthly residuals (thin lines are the monthly residual amplitudes) reduced by a GRACE climatology model (Equation (4)) and related formal errors for the standard RL06 solutions (black and red), and those with the a priori stochastic model applied (blue and yellow) in mm geoid height. (**Row 4**) Relative residual RMS change in % of the solutions with stochastic modeling compared to the standard RL06 solutions.

Applying such a stochastic model reduces the need for co-estimating empirical parameters, as done in RL06. In Figure 7 (right), the monthly GRACE solutions for January 2007 are shown comparing the RL06 solution with a RL06 solution, where the derived stochastic model has been applied and a RL06 solution with stochastic modeling, but without the co-estimation of any empirical parameters (once per revolution accelerations). From the SH degree amplitudes of the residuals, it can be seen that with stochastic modeling and without the empirical parameters, a similar noise level in the low and medium SH degrees can be reached as for RL06. Moreover, for high SH degrees the noise can be significantly reduced. Due to the reduced number of unknown parameters to be estimated, the formal errors' amplitudes are reduced when neglecting the empirical parameters.

Another performance parameter is the weighted ocean RMS of the residuals for different filters (cf. Table 2). For these filters, the high SH degrees are suppressed, and the differences in the low to medium SH degrees become more prominent. For Gaussian radii above 300 km, which emphasize the contribution of the very low SH degrees, applying the stochastic model degrades the RL06 performance. However, excluding the empirical parameters for these radii leads to the same performance level again. For Gaussian radii smaller than 350 km, the ocean RMS is reduced by nearly 10% by applying stochastic modeling and by more than 10% through additionally excluding empirical parameters.

**Table 2.** Weighted ocean RMS of surface mass densities in cm equivalent water height of the residuals of the three solutions for January 2007, shown in Figure 7 (right) for different Gaussian filter radii ($C_{20}$ coefficients neglected).

| Gauss. Radius in km | RL06 | RL06 w/stoch. Mod. | RL06 w/stoch. Mod. w/o empi.coef. |
|---|---|---|---|
| 500 | 1.4 | 1.9 | 1.4 |
| 450 | 1.6 | 2.0 | 1.6 |
| 400 | 2.0 | 2.3 | 2.0 |
| 350 | 2.8 | 3.0 | 2.7 |
| 300 | 4.8 | 4.7 | 4.5 |
| 250 | 10.7 | 9.7 | 9.3 |

## 5. Conclusions

In this study, we analyzed the key GRACE instruments, i.e., the ACC and the MWI, to assess their in-orbit performance. The methods applied for both instrument types aimed to cancel as much signal as possible from the observations to retain sensor noise exclusively. For the ACCs we transplanted the observations from one of the GRACE satellites to the other under consideration of all necessary corrections and subtracted these two data sets from one another. For the MWI we used the post-fit residuals obtained from monthly gravity field determination, applying GFZ GRACE RL06 processing standards.

The key results are noise ASDs and analytical models representing the three ACC axes and the MWI's stochastic behavior for the years 2007 and 2014. While the noise in all three ACC axes is shown to be higher than their pre-launch requirements, the opposite is the case for the MWI. Furthermore, the ACC noise level in 2014 is significantly increased in comparison to 2007, while the MWI noise level for frequencies above 10 mHz remains constant. The observed range rate post-fit residual ASDs for frequencies above 10 mHz are dominated by the MWI system noise. Besides other effects not related to instruments such as, e.g., background model errors, the along-track ACC component is one of the major noise contributors to range rate post-fit residuals for frequencies between 1 and 10 mHz.

In future, additional periods—especially from the GRACE mission's early years—shall be investigated to obtain a more complete picture of the ACC performance. As the performance differences between 2007 and 2014 are significant, the derived products can only be assumed as valid exclusively for these test years. It is, thus, of great value to understand whether the instrument noise levels feature long-term variations or whether there are any rapid fluctuations.

A combined ACC+MWI stochastic model, in the form of a filter matrix, is derived from the individual noise models obtained for the two instruments. Applying this combined stochastic model for the monthly solutions in 2007 and 2014 results in more realistic formal errors in comparison to those of the standard RL06 solutions, especially with regard to their relative behavior. The estimated gravity field parameters themselves, on the other hand, show a similar noise level for these two test years. Applying stochastic modeling only notably improves the gravity retrieval performance in the high-degree spectrum, as well as within certain low SH orders.

As our approach does not consider the stochastic properties of other error sources, such as background model errors or GPS observations, the results are not optimized with regard to the relative weighting of the different observation systems. Therefore, it is of great importance for future research that the stochastic behavior of all relevant components of the gravity adjustment process is modeled adequately. In this regard, important contributions have already been made in [37]. Furthermore, when treating the ACC observations as real observations within the adjustment process, the derived ACC stochastic models could be applied to the ACC observations in a more direct way. In addition to the presented application of an a priori stochastic model, one could further optimize the estimated parameters and their formal errors by iteratively applying an a posteriori model derived from range rate post-fit residuals.

**Author Contributions:** Conceptualization, M.M.; methodology, M.M. and P.A.; software, M.M., C.D. and P.A.; validation, M.M., C.D. and M.H.; writing—original draft preparation, M.M. and P.A.; writing—review and editing, M.M., P.A., C.D., M.H., R.P. and F.F.; visualization, M.M.; supervision, F.F. and R.P.; project administration, F.F. and R.P. All authors have read and agreed to the published version of the manuscript.

**Funding:** This work is funded by the Deutsche Forschungsgemeinschaft (DFG) within the research unit New Refined Observations of Climate Change from Spaceborne Gravity Missions (NEROGRAV, DFG Research Unit 2736). The mentioned aspects of future research are part of the second phase of this research unit.

**Data Availability Statement:** In https://doi.org/10.5880/nerograv.2023.001, the combined ACC+MWI stochastic models discussed in this paper, together with a brief data description, can be found.

**Conflicts of Interest:** The authors declare no conflict of interest.

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
