# Peer review of "In-Orbit Performance of the GRACE Accelerometers and Microwave Ranging Instrument"

_remotesensing, doi:10.3390/rs15030563_

Round 1

Reviewer 1 Report

The manuscript needs a minor revision and needs to be read by a native English-language spoken person.

The abbreviations should be mention in the first place of appearance and with capital letters. The following abbreviations should be corrected for capital letters. 

19: MicroWave ranging Instrument (MWI), ACCelerometer (ACC)

38: Terrestrial Water Storage (TWS)

47: Satellite-to-Satellite Tracking (SST)

53: Spherical Harmonics (SH)

187: Power Spectral Density (PSD)

187: Amplitude Spectral Density (ASD)

16, 36 : The GRACE sands for "Gravity Recovery And Climate Experiment"

41: "directly measuring changes in mass of the ice sheets": I would not call it "directly measuring".. computation of the L2 gravity field solutions and the further analysis of these products to make statements about mass changes of ice sheets is quite complex
63: "One potential field of research during gravity field estimation is the stochastic modeling of observation uncertainties.": is this the first work focusing on such a topic?... if not I suggest to include a short review of related articles
91: "Besides the gravity signal, the MWI data consists of sensor noise and other error effects...":
I suggest to paraphrase this sentence or to omit "error". After reading this sentence one can assume that all effects besides gravity and sensor noise are "error" effects.
For example I would not label non-gravitational effects as errors.
93: Other studies focusing on GRACE-FO inter-satellite ranging data are [28, 29].: Study [28] does not focus on GRACE-FO data!
93: only two studies in over two decades of GRACE and the follow on mission?
section 2.1.: is this the JPL standard approach? what is special about this approach?
143: "The sampling of GNV1B and SCA1B is increased by means of linear interpolation to..." why linear interpolation?
154: "In order to remove this error source from the ACC1B data, all data points in the range of 40 s to both sides of..." so the filtering of L1A always results in a "symmetric" smearing?
equation 1: t and omega not defined
213: so only the a priori static gravity field is reduced from the L1B data? I would assume, that the prefit residuals would also not contain other effects (tides, relativistics,...) used for orbit propagation
figure 1: what is the purpose of the right panel?
249: "It can be shown that the noise level of each..." shown --> seen?
456: "(Row 3) Median of SH degree amplitudes (thick lines) of the monthly residuals (thin lines are the..." thin lines are hardly visible
490: "Error! Reference source not found". It should be compiled again.
496: "...to cancel as much as signal as possible..." --> as much signal as possible

539: The link is not valid and the corresponding ACC+MWI data can not be accessed.

The references could be expanded to cover more publications on classical processing methods for GRACE and GRACE-FO datasets.

Author Response

Dear Reviewer,

thank you very much for your helpful suggestions and questions. Please find attached our responses to your points.

Kind regards

Authors

Reviewer 2 Report

Dear colleagues,

The subject of the paper is very interesting and definitely worth publishing. I have the following remarks and I hope you find them useful and constructive. 

1. I have uploaded an annotated file with my comments which are mainly to the text. 

2. I think this paper is more suitable for MDPI Sensors than Remote Sensing but let us leave this to the editor to decide. 

3. The text is not very understandable, even if it is correct grammatically. The main point is to understand what the authors mean by sentences. There are many long sentences, which are hard to understand. The authors should put themselves in the position of readers who are probably not as deep as the authors are in the subject of the paper. There are in appropriate or less appropriate words, which are misleading. The use of parentheses in the sentences made really the text difficult to understand. In many places, the authors use specially terminologies without explaining them and probably they expect that readers should know them before reading the paper. I highly recommend rewriting the whole text in a simpler form. 

3. The methodology of stochastic modelling was not presented properly, giving only an equation is not enough. This part of the paper needs a significant improvement. 

4. There are arguments which needs justifications; see the annotated file. 

5. A statistical test needs to be done for justifying the improvements. Plotting the degree variances and plotting the RMSs, which are close in similar scenarios of filtering over ocean is not convincing. 

Generally, I support the idea of the paper, but not the paper itself. It needs a major revision in my opinion, to make the paper publishable. 

I wish you good luck with the revision.

Best regards

Reviewer.

Author Response

Dear reviewer,

thank you very much for your suggestions and questions. Please find attached our responses to your points.

Kind regards

Authors

Reviewer 3 Report

The manuscript developed realistic stochastic models for two key onboard GRACE instruments ACC and MWI by analyzing L1B data residuals in 2007 and 2014. Their results show that the in-orbit performance of ACC is worse than its pre-launch specification while that of the MWI is better than the expectation. The innovation of the manuscript is the adoption of a ‘transplant approach’ to obtain the ACC1B data residuals, which were not easily accessible in previous studies as we generally do not treat ACC1B data as ‘real’ observations during the parameter estimation process for GRACE gravity field recovery. Using the combined ACC+MWI stochastic model, the gravity field solutions are improved for high-degree SHC, and more realistic formal error estimates are obtained, which can be very important for subsequent model combinations.In summary, the manuscript is well-written, and I recommend it for publication provided the following issues are addressed.

Major issues

1.The ACC noise is approximated by the transplant residuals between two GRACE satellites based on the assumption that the time-shifted ACC1B data represent the same non-gravitational signals. However, from my understanding, subtracting the homogenized ACC1B data from one another will cancel out the common part of the ACC noise (if we assume the two accelerometers are of very similar instrument performances), which can result in an underestimate of the noise level. If this is the case, it should be addressed in the manuscript to indicate the possible drawbacks of the ‘transplant approach’ for approximating the ACC noise;

2.As the title writes ‘in-orbit performance of the GRACE …’, the results of only two years 2007 and 2014 as shown in the manuscript may not adequately represent the whole period of GRACE, at least the results for early years, e.g., before 2005, are missed, for which we would expect degraded performances with respect to 2007 and 2014? But maybe just a few remarks/discussions on this at the end of the manuscript will address my concern.

Minor remarks

1.Page 3, line 134: please indicate here which version of L1B data is used. The GRACE L1B RL03 provides 1-second SCA1B data;

2.Page 4, Eq. (1): I would suggest writing the formula in terms of the ‘observation equation’ of LS adjustment, and indicating which parameters are actually estimated for a better understanding. If I understood correctly, it is the differential parameters ‘delta_X’ but not the absolute parameters ‘X_A/B’ that are estimated (X stands for S, Q, D, 1/2-rev). Also, please indicate how the rotation matrix R_A/B is derived, e.g, SRF-B  CRF  SRF-A?

3.Page 12, lines 423-425: for the 3-hour filter length case, is the VCM used still of 24-hour dimension/size? If so, by replacing other elements apart from the 3-hour diagonal band with zeros, will the resulting 24-hour VCM still be positive-definite? 

Author Response

(The authors gave the same response as above.)

Round 2

Reviewer 2 Report

Dear Colleagues, 

Thank you very much for your collaboration in revising the manuscript. I am happy with your responses and revision. I recommend accepting this paper to the handling editor. 

Good luck with your paper!

Best regards

Mehdi Eshagh.